# Defining Wound Healing Progression in Cetacean Skin: Characteristics of Full-Thickness Wound Healing in Fraser’s Dolphins (*Lagenodelphis hosei*)

**DOI:** 10.3390/ani12050537

**Published:** 2022-02-22

**Authors:** Chen-Yi Su, Michael W. Hughes, Tzu-Yu Liu, Cheng-Ming Chuong, Hao-Ven Wang, Wei-Cheng Yang

**Affiliations:** 1School of Veterinary Medicine, National Taiwan University, Taipei 10617, Taiwan; angelsu2096@gmail.com; 2International Center for Wound Repair and Regeneration, National Cheng Kung University, Tainan 701, Taiwan; mwhughes@usc.edu (M.W.H.); brothansis@yahoo.com.tw (T.-Y.L.); 3Institute of Clinical Medicine, National Cheng Kung University, Tainan 701, Taiwan; 4Department of Life Sciences, National Cheng Kung University, Tainan 701, Taiwan; 5Department of Pathology, Keck School of Medicine, University of Southern California, Los Angeles, CA 90033, USA; cmchuong@med.usc.edu; 6Marine Biology and Cetacean Research Center, National Cheng Kung University, Tainan 701, Taiwan

**Keywords:** dolphins, wound healing, rete ridges, adipose tissue, melanocytes, regenerative medicine

## Abstract

**Simple Summary:**

Cutaneous wound healing is a complex and tightly regulated biological process to restore physiological and anatomic function. Current knowledge of cutaneous wound healing is mostly based on studies in laboratory animals and humans. The histological and immunological features of skin, for example, cutaneous thickness, cellular components, and immune response, are not identical among animal species, and these differences may lead to substantial effects in cutaneous wound healing. In field observation, large cutaneous wounds in cetaceans could heal without medical treatments. However, little is known about the underlying mechanisms, and there is no histological study on full-thickness wound healing in cetaceans. The current study characterizes the macroscopic and histological features of large full-thickness wound healing in Fraser’s dolphins (*Lagenodelphis hosei*). The differences of wound healing between cetaceans and terrestrial mammals were shown from the histological aspect, including rete and dermal ridge appearance, repigmentation, and adipose tissue regeneration. Better understanding of the mechanism of full-thickness wound healing in cetaceans will shed light on veterinary and human regenerative medicine, leading to novel therapies.

**Abstract:**

Cetaceans are tight-skinned mammals that exhibit an extraordinary capacity to heal deep soft tissue injuries. However, essential information of large full-thickness wound healing in cetaceans is still lacking. Here, the stages of full-thickness wound healing were characterized in Fraser’s dolphins (*Lagenodelphis hosei*). The skin samples were collected from normal skin and full-thickness cookiecutter shark (*Isistius brasiliensis*)-bite wounds of stranded carcasses. We defined five stages of wound healing according to macroscopic and histopathological examinations. Wounds in Stage 1 and 2 were characterized by intercellular and intracellular edema in the epidermal cells near the wound edge, mixed inflammatory cell infiltration, and degradation of collagen fibers. In Stage 3 wounds, melanocytes, melanin granules, rete and dermal ridges were noticed in the neo-epidermis, and the adipose tissue in adjacent blubber was replaced by cells and fibers. Wounds in Stage 4 and 5 were characterized by gradual restoration of the normal skin architecture including rete and dermal ridges, collagen bundles, and adipose tissue. These phenomena were quite different from previous studies in terrestrial tight-skinned mammals, and therefore, further in-depth research into the mechanisms of dolphin wound healing would be needed to gain new insights into veterinary and human regenerative medicine.

## 1. Introduction

In humans and most mammals, cutaneous wound healing is a complex biological process which has been divided into three continuous and overlapping phases: inflammation, proliferation, and remodeling [1]. The inflammation phase occurs first and directly after skin injury. A fibrin clot forms immediately to re-establish hemostasis and prevent invasion of microorganisms as well as providing a scaffold for inflammatory and other cells to crawl into the wounded area [2]. Inflammatory cells, such as neutrophils and macrophages, are recruited to the wounded area for clearing invading bacteria and cellular debris. As the clearing process approaches an end, a change of local microenvironmental signals promotes macrophage phenotype switching from a pro-inflammatory phenotype to an anti-inflammatory phenotype, contributing to the progression of wound healing [3,4]. The second phase of wound healing is the proliferation phase, which includes angiogenesis, granulation tissue formation, and re-epithelialization. Granulation tissue is composed of primarily new blood vessels, immune cells, fibroblasts, and an abundance of extracellular matrix (ECM) produced by the fibroblasts [5]. Re-epithelialization, which involves the proliferation, differentiation, and migration of keratinocytes, is an essential process for wound closure [6]. Furthermore, wound contraction promoted by myofibroblasts aids wound closure [7]. In the last phase of wound healing, most of the cells in the granulation tissue undergo apoptosis or emigrate from the wounded area, leaving a scar tissue containing few cells with an excess of abnormal ECM [2]. ECM composition changes during wound healing processes are under the regulation of cytokines, growth factors, and matrix metalloproteinase (MMPs). These are secreted by fibroblasts, macrophages, and endothelial cells [8]. ECM remodeling enhances the strength of repaired tissue; however, the repaired tissue does not return to a completely normal status in humans and most tight-skinned mammals [9].

Although the basic mechanisms of wound healing are preserved in most mammals, heterogeneity in wound healing has been found in different species, individuals, and anatomical sites [10,11,12,13]. For example, wound closure in tight-skinned species, such as humans and pigs, is primarily by re-epithelialization, whereas wound closure in loose-skinned animals, such as rats, mice, and rabbits, is primarily caused by contraction of the panniculus carnosus muscle (reviewed in [12]). Another example is the rate of granulation tissue formation and wound contraction in cats being rather slow compared to dogs, even though both dogs and cats are closely related in the order Carnivora. This difference could be related to variability in cutaneous blood supply during wound healing [10]. 

Heterogeneity in wound healing is not a phenomenon restricted to between species but also exists within the same species. Horses and ponies, both belonging to *Equus ferus caballus*, show considerable differences in second-intention wound healing [13]. The authors noted that the inflammatory response in ponies was strong but short, leading to a better quality second-intention healing, while horses had a weak and chronic inflammatory response, resulting in the production of poorly vascularized exuberant granulation tissue. Previous studies showed that the formation of hypertrophic scars after wounding can be observed in red Duroc pigs and Mexican hairless dogs but not in other breeds of pigs and dogs [14,15,16]. Genetic variations leading to heterogeneity in wound healing also occur in humans. The prevalence of keloid scarring in certain ethnic populations, such as African American, Asian, Mediterranean, and Hispanic, is higher than in Caucasian populations [11,17]. 

Although the heterogeneity of wound healing in several terrestrial mammals has been studied, little is known about the timing, sequence, and mechanism of wound healing in mammals that live in a significantly different environment: the marine habitat. Cetaceans, which inhabit water for the entirety of their lives, have been reported to possess great healing capacity [18,19,20]. Cetaceans possess a unique cutaneous anatomy, which is supposed to be an adaptation to the aquatic environment [21]. Cetacean skin is characterized by an exceptionally smooth, rubbery texture without hairs or other skin appendages (excepting newborn odontocetes that possess hairs on the rostrum and mysticetes that have vibrissae on the rostrum and mandible) [22,23]. The epidermis consists of a stratum externum, a stratum spinosum, and a stratum basale; the stratum granulosum and stratum lucidum are absent [22]. Melanin granules are distributed in all epidermal layers, including the stratum externum [24]. Most of the cells in the stratum externum retain flattened nuclei, which is referred to as parakeratosis [22]. A thick epidermis with elongated rete ridges interdigitating with dermal ridges (dermal papillae) is one of the extraordinary characteristics of cetacean skin. The serrated interface between epidermis and dermis increases the ratio of germinative to superficial cells, consequently contributing to a greater proliferative capacity in cetacean skin [25]. Moreover, the well-developed rete ridges could function to enhance skin adherence against hydrodynamic friction, which prevents the epidermis being torn off from the dermis during high-speed swimming [26]. Blood vessels and nerve fibers are present in the dermis and dermal ridges, providing nutrition and sensory perception [27]. Beneath the dermis is blubber, a thick layer of specialized subcutaneous adipose tissue found only in marine mammals [28]. Blubber contains numerous adipocytes, intermingled with blood vessels, nerves, collagen fibers, and elastin fibers [27,29]. The literature shows that blubber plays an important role in thermoregulation, metabolic energy storage, buoyancy, body streamlining, and locomotion [30].

Cetaceans commonly sustain traumatic injuries such as abrasions, conspecific biting or shark biting, anthropogenic trauma caused by fishing entanglement or propeller strike in free-ranging individuals, and thermal burns in stranded cetaceans [18,31,32,33,34,35]. Interestingly, observations report deep cutaneous wounds in wild cetaceans can completely heal without medical treatment [20]. Some antimicrobial compounds, for example organohalogens and isovaleric acid, were found in the blubber layer [36,37] and could provide a natural protection against infection contributing to wound healing [20]. Furthermore, the regenerative ability in cetacean skin is associated with the adipose-derived stem cells existing in the blubber layer [38]. 

To the best of our knowledge, there are only two studies describing the histological features of cutaneous wound healing in cetaceans [39,40] that focused on superficial wound healing in bottlenose dolphins and beluga whales, respectively. The studies showed that both species shared a similar sequence of cellular and vascular change during the healing process with humans and laboratory animals. The only remarkable difference between cetaceans and terrestrial mammals was no solid fibrin clot or scab formation during wound healing in cetaceans. Instead, a layer of degenerative cells mixed with vesicles covered the wound surface in cetaceans. The authors suggested that the formation of this degenerative layer might relate to the change of cellular osmolarity in cells when exposed to seawater. This could provide a similar function to a solid fibrin clot or scab in terrestrial mammals to be a protective barrier between the external environment and the underlying tissue.

Although the aforementioned studies on superficial wound healing in captive dolphins and whales described the sequence and the timing of wound healing, the essential information of large full-thickness wound healing in cetaceans is still lacking. The aim of the current study was to describe the macroscopic and histological features of full-thickness cutaneous wound healing in Fraser’s dolphins (*Lagenodelphis hosei*). It would not be surprising if there are significant differences in large full-thickness wound healing between cetaceans and other mammals. Understanding the differences and underlying mechanisms may contribute to the development of novel therapies to treat severe trauma patients (both non-human and human) and make progress in the field of skin wound healing research.

## 2. Materials and Methods

### 2.1. Sample Collection

The skin samples used in the current study were collected from four dead stranded Fraser’s dolphins. The samples included normal dolphin skin and full-thickness wounds caused by cookiecutter shark (*Isistius brasiliensis*) bites. These shark bite wounds were approximately 5–8 cm in diameter and 2–3 cm in depth, and mainly present on the dorsolateral and ventral parts of the body. Wounds in different healing states, including recently created, healing, and healed, were collected for subsequent analysis. Samples in poor condition (e.g., sloughing epidermis) were ruled out from the current study. Body condition and carcass condition were assessed according to previous studies [41,42]. Freshly dead animals were classified into carcass Code 2; animals presenting moderate decomposition were classified into carcass Code 3. The sample condition and the details of each animal are listed in Table 1.

### 2.2. Tissue Preparation and Histochemical Staining

The skin samples taken from the stranded dolphins were fixed in 10% neutral buffered formalin for 3 days and then embedded in paraffin. For hematoxylin and eosin (H&E) staining and Fontana-Masson staining, the tissues were cut into 4 μm sections; for Masson’s trichrome staining, the tissues were cut into 5 μm sections. Slides were deparaffinized in xylene and rehydrated in graded ethanol, and then subjected to H&E staining, Fontana-Masson staining, and Masson’s trichrome staining according to accepted protocol. Images were recorded with a Whited WM100 microscopy (Whited, Taipei, Taiwan).

## 3. Results

### 3.1. Normal Skin

The epidermis consisted of a stratum externum, stratum spinosum, and stratum basale. No hair follicles or other skin appendages were observed. The outermost layer of the epidermis was composed of flattened keratinocytes with pyknotic nuclei, defined as parakeratosis. Some of the spinous cells showed clear nuclear haloes. Melanin granules appeared in all layers of the epidermis. Prominent epidermal rete ridges extended downward into the dermis to construct an interdigitated interface between epidermis and dermis (Figure 1). The dermis consisted of a complex network of collagen fibers. Blood vessels and nerve fibers were clearly observed in the reticular dermis. The blubber layer was rich in adipocytes, intermingled with collagen fibers, elastin fibers, nerve fibers, and blood vessels. Collagen fibers presented in a basket-weave orientation. A higher quantity of blood vessels was observed in the deep layer of blubber than in the superficial layer.

### 3.2. Wounds

Five stages were classified in the wound healing progress in Fraser’s dolphins (Table 2 and Figure 2): Stage 1, new wound; Stage 2, initially healing wound without granulation tissue; Stage 3, healing wound with granulation; Stage 4, healed wound with cellular and vascular blubber; Stage 5, healed wound without cellular and vascular blubber.

#### 3.2.1. Recently Created or Immature Wound; Stage 1

Grossly, the wounds showed sharp edges with little or no bleeding. Underlying tissues were exposed to the environment with no necrotic tissue covering. Histologically, intercellular and intracellular edema were noted in the stratum spinosum near the edge of the wound (Figure 3). In the edematous region, the epidermis was pale and swollen. Some keratinocytes presented with karyolytic, pyknotic, or karyorrhectic nuclei while some presented without nuclei (Figure 3B). There was no obvious change in the dermis and blubber. Only a few collagen fibers at the wound edge were necrotic. This stage of cetacean wound healing was classified as Stage 1.

#### 3.2.2. Initiation of Wound Healing in Wound; Stage 2

The macroscopic appearance of the wounds was characterized by a raised and puckered epidermal edge with a layer of yellow-white substance covering the surface of the wounds. Histologically, the epidermal cells near the wound edge were edematous and the region of cellular edema extended to the adjacent area, approximately 2.4–7.9 mm from the wound edge (Figure 4B). Exfoliation of degenerating epidermal cells was noted. The yellow-white substance covering on the wound surface consisted of necrotic collagen and necrotic adipose tissue, approximately 0.3–4.2 mm in width (Figure 4C). Next to the necrotic tissue was an inflammatory cell infiltration zone, approximately 0.7–6.3 mm in width (Figure 4D). The predominant inflammatory cells were granulocytes. Adjacent to the inflammatory cell infiltration zone, hyperemia and hemorrhage were noted in the blubber, predominantly in the middle and lowest layers of blubber, extending up approximately 10 mm on either side of the wound edge (Figure 4E). Degradation of collagen fibers in the dermis and blubber close to the wound edge was noted (Figure 5A,B). Adipose tissue in the blubber was replaced by a loose fibrin network composed of fibrin, erythrocytes, and inflammatory cells (Figure 5C,D). This stage of wound healing was classified as Stage 2.

#### 3.2.3. Mature Open Healing Wound; Stage 3

Under gross view, neo-epidermis showed in the wound margin. In the center of wound, the void was filled with a reddish tissue. Histologically, rete and dermal ridges were noticed in the neo-epidermis (Figure 6B). Melanocytes and melanin granules were also observed in the neo-epidermis (Figure 6C). Part of the thick collagen fibers broke into pieces and part of the adipose tissue was replaced by numerous cells and thin fibers (Figure 6D). In the wound center, reddish granulation tissue consisted of cells, extracellular matrix, and numerous microscopic blood vessels (Figure 6E). Trichrome staining showed that thin collagen fibers and blood vessels appeared in the papillary dermis that connected to the neo-epidermis (Figure 6F,G). Keratinocytes in the neo-epidermis and uninjured adjacent skin were enlarged and edematous, approximately extending 7.8 mm away from the wound edge. In the uninjured area adjacent to the wound, the degenerating epidermal cells had exfoliated, resulting in the loss of the stratum externum and part of the stratum spinosum. This stage of wound healing was classified as Stage 3.

#### 3.2.4. Immature Healed Wound; Stage 4

Grossly, the surface of wound was fully epithelialized. Histologically, no cellular edema was observed in the epidermis. Melanin granules appeared in all the layers of neo-epidermis. Rete and dermal ridges were present in the neo-epidermis. According to the histological features, such as collagen thickness, cell density, and blood vessel density, this category of healed wounds was subdivided into two stages. The dermis and blubber layer were filled with numerous thin collagen fibers which oriented parallel to the skin surface. Numerous blood vessels, fibrocytes, fibroblasts, and other cells existed among the collagen fibers (Figure 7B). Thick collagen fibers were present in the lower part of the blubber (Figure 7C). Scattered rounded spaces were noticed beside blood vessels. Nerve fibers were rarely observed. A wound with the features mentioned above was classified as Stage 4.

#### 3.2.5. Mature Healed Wound; Stage 5

In macroscopic view, the mature healed wound was closed, showed little to no contraction lines, and was evenly pigmented similar to the surrounding unwounded skin. In the more “mature” wounds, which were classified as Stage 5, the dermis and blubber layers were less cellular and vascular (Figure 8). The presence of rete and dermal ridges was observed (Figure 8B). It was clear that adipocytes appeared in the perivascular region (Figure 8C), but the amount of adipose tissue was varied among different samples. Most of the wounds in this stage comprised thin collagen fibers, while a few contained thick collagen fibers similar to the collagen bundles in the normal skin. Few nerve fibers could be observed in the dermis. The architecture of Stage 5 healed wounds was significantly similar to unwounded skin.

## 4. Discussion

In the current study, the process of full-thickness wound healing in Fraser’s dolphins was characterized. The normal skin structure of Fraser’s dolphins is quite similar to previously described cetacean species [22,27]. The skin characteristics of these cetacean species share several similarities, for example, parakeratosis in the outermost layer of the epidermis, absence of hair follicles and other skin appendages, relatively simplified cellular strata in the epidermis, prominent epidermal rete and dermal ridges, and the distribution of blood vessels in the dermis and blubber. Compared to terrestrial tight-skinned mammals, humans and pigs, the thickness of normal cetacean epidermis is much greater. Increased thickness of the epidermis happens in humans, but only during abnormal conditions, for example, injury or autoimmune diseases like psoriasis [43]. A thickened epidermis in cetaceans could provide better protection against environmental insults and maintain homeostasis in the aquatic environment and is considered one adaptation to the aquatic life [21,25,44]. The current study showed the presence of rete and dermal ridges after full-thickness wounding indicating the importance of rete ridges in dolphin skin. These structures do not appear after full-thickness wound healing in humans and pigs, and only limited rete and dermal ridge regeneration was observed in partial-thickness wounds in pigs [14]. The interdigitated rete and dermal ridges create an epidermis firmly connected to the dermis, preventing detachment from the underlying tissue during high-speed swimming [26]. It was surprising to observe rete and dermal ridges in the neo-epidermis in Stage 3 wounds in the current study, which might provide better adhesion during the wound closure process. Moreover, the convoluted basement membrane is accompanied by a higher ratio of basal cells to superficial cells compared to terrestrial mammals, consequently increasing the proliferative capacity and potentially contributing to wound closure during skin wound healing [39,45].

The epidermis layer in Stage 1 and Stage 2 was edematous without a solid fibrin clot or scab, similar to reports for superficial wounds in bottlenose dolphins and beluga whales [39,40]. A previous field study showed that the surface of shark-inflicted wounds was covered with blubber coming from the wound adjacent area within the first day after injury [20]. However, our findings were not consistent with this. The macroscopic appearance of the wounds in Fraser’s dolphins was a layer of yellow-white substance covering the surface of the wounds in Stage 2, and this substance was necrotic collagen and adipose tissue in the microscopic observation. A layer of necrotic tissue might serve as a mechanical barrier to maintain homeostasis and protect underlying tissues from further damage and enable wound healing. 

Of note, regions of adipose tissue in the blubber were replaced by a loose fibrin network that could provide a scaffold for the migration of fibroblasts, inflammatory cells, and other cell types to the wounded area [9]. Recent studies have shown that mouse adipocytes can undergo cellular reprogramming by dedifferentiation into preadipocytes and mesenchymal stem cells, or by transdifferentiation into myofibroblasts, indicating that adipocytes have a high cellular plasticity (reviewed in [46]). The disappearance of adipocytes during wound healing in cetaceans might result from apoptosis, or dedifferentiation or transdifferentiation into other cells. Cetaceans possess a thick layer of subcutaneous adipose tissue. This raises the question of if the adipose tissue around the full-thickness wound in cetaceans supplies essential fibroblasts and endothelial cells utilized for the formation of granulation tissue during wound healing. Granulation tissue consists of primarily new blood vessels, immune cells, fibroblasts, and an abundance of extracellular matrix (ECM) [5]. In full-thickness wounds, the void of the wound needs to be filled with granulation tissue before re-epithelialization [7]. A full-thickness cookiecutter shark wound in a human required a significantly long time to heal, even though a skin graft had been applied [47]. In contrast, a similar wound on a stranded cetacean was filled with granulation tissue within days, followed by epithelial migration, and completely closed in the next two months (H.-V.W., personal communication). Previous studies showed the origin of fibroblasts forming the granulation tissue in full-thickness wounds could arise from multiple populations of cells, for example, perivascular sheaths and pericytes (shown in rabbits), hematopoietic cells with mesenchymal characteristics (fibrocytes) (shown in mice), and stromal elements in the adipose layer (shown in rabbits and pigs) (reviewed in [7]). If adipose tissue in cetacean skin is an essential element to supply fibroblasts and endothelial cells for the formation of granulation tissue, adipose tissue regeneration would be necessary in order to prepare required materials for future wounds. Further studies are required to understand the mechanism of the rapid formation of granulation tissue in cetacean skin wounds.

Interestingly, small, rounded spaces with obvious membrane structures were noticed next to blood vessels in Stage 4 and Stage 5. We hypothesized that these small, rounded spaces could be neo-adipocytes and that this is the location of adipocyte regeneration after full-thickness wounding in cetaceans. Adipocyte regeneration has been studied in laboratory animals but is yet to be fully elucidated. In mouse studies, it was reported that adipogenic progenitors reside in the perivascular niche of adipose tissue, suggesting adipose progenitors might come from endothelial cells or pericytes (reviewed in [48,49,50]). Morphological and genetic evidence showed white and brown fat depots originated from cells which displayed endothelial characteristics [51]. The authors further suggested the possibility of cellular reprogramming with the interconversion between adipocytes and endothelial cells, contributing to the maintenance of homeostatic equilibrium during adipose tissue expansion and reduction. It would be interesting to investigate the cellular reprogramming in the possible coordination between angiogenesis in Stage 3 and adipogenesis in Stage 4 in wound healing process in cetaceans.

Wound healing with less fibrosis has been reported in humans under certain circumstances, for example, adult oral mucosa wounds and early gestation fetal skin wounds. These wounds exhibit less inflammatory cell infiltration during wound healing, which is considered one of the critical factors for scarless wound healing [52,53]. Of note, inflammatory response occurred in a comparatively smaller area during the healing process in large full-thickness wounds of Fraser’s dolphins. It needs to be emphasized that the cookiecutter shark bite wounds collected in this study were relatively large, 5–8 cm in diameter and 2–3 cm in depth. Similar wounding in humans requires medical intervention and more than six months to repair [47]. However, the infiltration of inflammatory cells in Fraser’s dolphin skin only extended approximately 4–7 mm away from the edge of a Stage 2 wound. One of the factors contributing to the limited inflammatory response might be certain special compounds in cetacean skin. It was reported that blubber contains organohalogens and a high proportion of isovaleric acid, which exhibit antimicrobial properties [36,37,54]. The presence of these antimicrobial compounds in blubber could provide a synergistic effect to defend the body against infection and restrict pathogen distribution to a relatively small area. Another possible explanation for this phenomenon is that the inflammatory response in cetaceans might be able to eradicate pathogens more efficiently than other animals. Although the inflammatory phase is generally believed to be vital for wound healing to proceed, there is evidence to suggest otherwise [8]. Skin wounds in PU.1 null mice, which lacked macrophages and functioning neutrophils, could heal within the normal time course [55]. Moreover, wounds in these macrophage-removed mice healed without scar, similar to what is reported for fetuses. This study demonstrated that inflammatory cells might not be essential in wound healing as long as microbial infection is controlled. However, it has been reported that the metabolic regulation of innate immune cell phenotypes is significant during skin regeneration, regeneration-competent versus -incompetent mice differ in neutrophil ability, and macrophages are required for skin regeneration in African spiny mice [56,57,58]. There is a lack of information on the immune response during full-thickness wound healing in cetacean skin. Further studies on the relationship between immune modulation and the enhanced healing ability of cetacean skin is essential. Moreover, it could be hypothesized that bacterial infection during wound healing in cetaceans may hinder skin regeneration and lead to impaired wound healing and excessive scarring.

Melanin in the skin is synthesized by melanocytes and is suggested to provide protection of epidermal cells from ultraviolet radiation (UVR)-induced DNA damage [59]. A previous study showed that whales with more pigmentation possessed fewer UVR-induced skin lesions, indicating the importance of a photoprotective function for melanin in cetaceans [60]. Melanin was observed in all layers of the epidermis in cetaceans and the wide distribution of melanin was suggested to be a unique photoprotective strategy for the adaptation to aquatic life [24]. In addition to the photoprotective function, many studies showed the antimicrobial and immunomodulatory properties of melanin (reviewed in [59]). In the current study, melanocytes and melanin were observed in the migrating epithelial tongue in Fraser’s dolphins. This phenomenon was quite different from a previous study in pigs, which did not find melanin in the neo-epidermis of full-thickness wounds until they were fully re-epithelialized [61]. Studies on human melanocytes showed that they are not only a source for melanin but also participate in skin immunity through cytokine production, antigen recognition, and antigen presentation [59,62,63]. It remains unknown what the dynamic change of melanocytes during wound healing is, the relationship between intact and healed skin color, and the number of melanocytes in cetaceans. Furthermore, it is important to study the immune functions of melanin and melanocytes in cetacean skin and compare the findings with those in terrestrial mammals.

Differences of collagen bundle thickness, spacing, and orientation were observed at different healing stages in the current study. Thin collagen fibers in a parallel arrangement mixing with numerous fibroblasts, fibrocytes, and blood vessels were noticed in the dermis and blubber in Stage 4, while the collagen fibers were comparatively thicker, and the number of cells decreased in Stage 5. Previous studies showed that collagen thickness is related to the composition [64]. In normal human skin, type I and type III collagen are the two major types of collagens [8]. Type I collagen, which is relatively thicker, helps maintain skin structure and tissue integrity, while type III, which is relatively thin, provides tensility, flexibility, and softness [8,64]. It has been reported that normal human skin contains 80–90% type I collagen and 10–20% type III collagen [65]. During wound healing, type III collagen appears earlier than type I collagen, and the percentage of type III collagen may increase up to 30% in granulation tissue [8,66]. In the remodeling phase, type III collagen undergoes degradation, and the proportion of the two collagen subtypes gradually returns to approximately normal ratios in the mature scar [8,66]. Cetaceans can dive several hundreds to thousands of meters deep [67]. Collagens in cetacean skin are essential to maintain skin architecture and body profile during diving. Cetaceans’ diving capacity could be compromised if healed wounds did not function similar to normal skin, especially for a cetacean with numerous healed wounds. However, little is known about the composition of collagen content in cetacean skin, and it would be interesting to understand if cetacean healed wounds equip a normal composition of collagen. Further studies on the quantification of different collagen subtypes in cetacean skin is needed. We surmise that the composition of collagens in cetacean skin, and the timing of ECM remodeling, is different from humans due to the exposure to an extreme marine environment and high water pressure during diving.

## 5. Conclusions

To the best of our knowledge, this is the first study to characterize histological features of full-thickness wound healing in cetaceans, which demonstrate this adaptation to the aquatic environment. The most significant findings in cetacean full-thickness wound healing included (1) the early appearance of melanocytes and melanin during wound healing; (2) the presence of adipose tissue and rete and dermal ridges in the completely healed wound. These phenomena are quite different from those in terrestrial tight-skinned mammals. It shows that the full-thickness wounds in cetaceans heal in a regenerative manner rather than repair. We hypothesize that the thick blubber layer and limited inflammatory response are critical factors contributing to the enhanced healing ability of cetacean skin. Further studies to elucidate the mechanisms of immune modulation, angiogenesis, adipocyte regeneration, and collagen reconstruction during full-thickness wound healing in cetaceans may shed light on veterinary and human regenerative medicine, leading to novel therapies.

## Figures and Tables

**Figure 1 animals-12-00537-f001:**
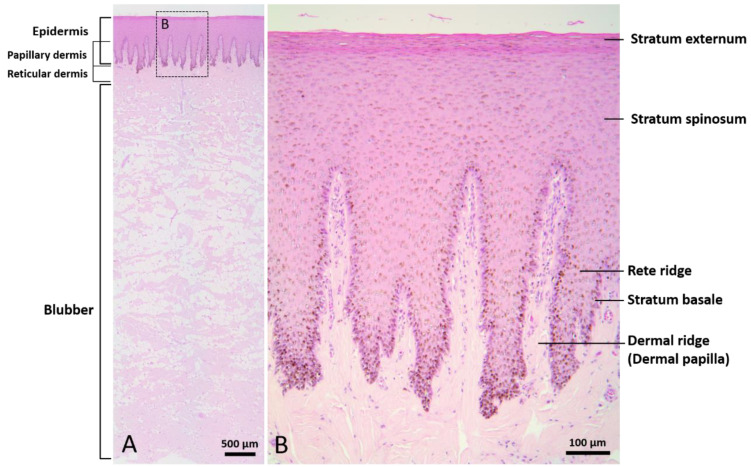
Normal skin of Fraser’s dolphin. H&E staining of Fraser’s dolphin normal skin. (**A**) Low power view of epidermis, dermis, and blubber. Scale bar = 500 µm. (**B**) High power view of epidermis and dermis. Flattened cells with pyknotic nuclei were observed in the stratum externum. Melanin granules were distributed in all layers of the epidermis. Prominent rete ridges and dermal papillae formed an interdigitated interface between epidermis and dermis. Scale bar = 100 µm.

**Figure 2 animals-12-00537-f002:**
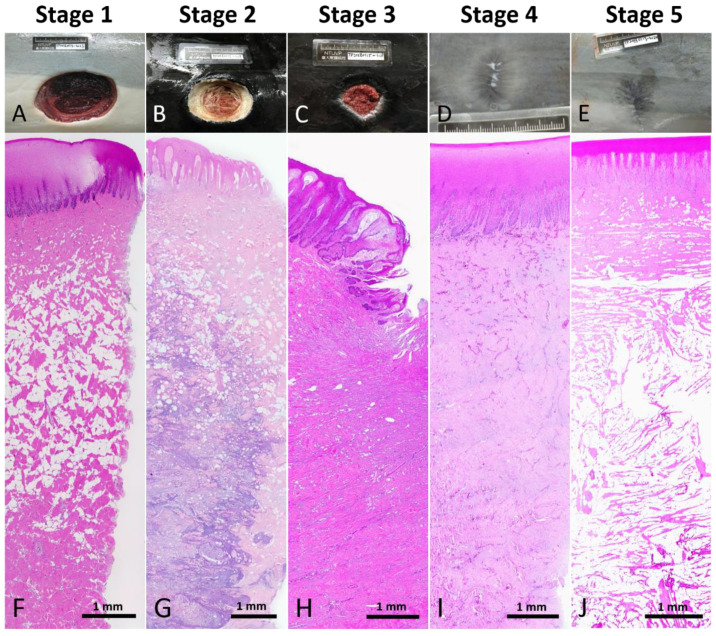
Reference macroscopic and low power microscopic images of each wound healing stage. (**A**,**F**) Stage 1 wound. (**B**,**G**) Stage 2 wound. (**C**,**H**) Stage 3 wound. (**D**,**I**) Stage 4 wound. (**E**,**J**) Stage 5 wound.

**Figure 3 animals-12-00537-f003:**
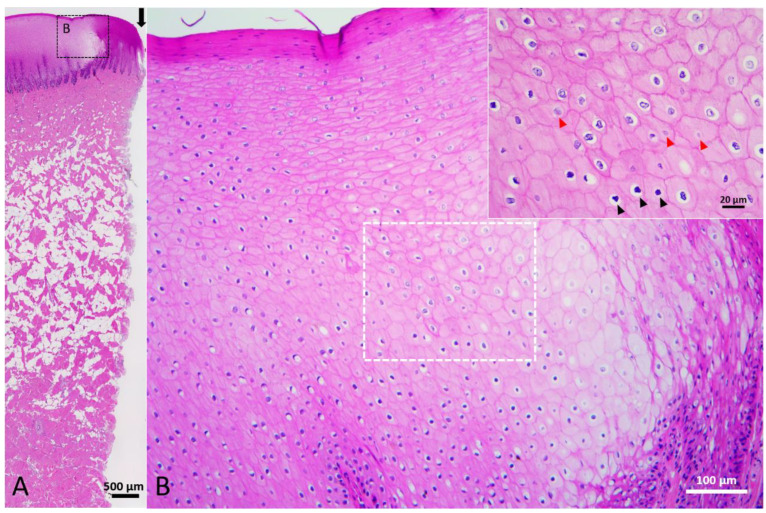
Stage 1 wound of Fraser’s dolphin. H&E staining of Fraser’s dolphin Stage 1 wound. (**A**) Low power view of Stage 1 wound. There was no obvious change in the dermis and blubber. A few collagen fibers at the wound edge were necrotic. Black arrow: edge of wound. Scale bar = 500 µm. (**B**) High power view of Stage 1 wound epidermis. Epidermal cells were pale, enlarged and edematous. Karyolysis (red arrowhead) and pyknosis (black arrowhead) were observed. Scale bar = 100 µm.

**Figure 4 animals-12-00537-f004:**
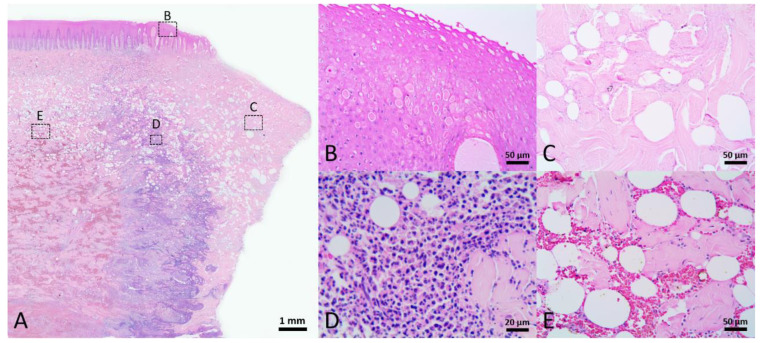
Stage 2 wound of Fraser’s dolphin. H&E staining of Fraser’s dolphin Stage 2 wound. (**A**) Low power view of Stage 2 wound. Scale bar = 1 mm. (**B**) High power view of wound edge epidermis. Near the wound edge, keratinocytes were edematous with pyknotic nuclei or even without nuclei. Scale bar = 50 µm. (**C**) High power view of blubber on the wound surface. The collagen and adipose tissue were necrotic. Scale bar = 50 µm. (**D**) High power view of blubber nearby the wound edge. Inflammatory cells infiltrated in the blubber. Scale bar = 20 µm. (**E**) High power view of blubber away from the wound edge. Hyperemia and hemorrhage occurred in the blubber, mainly in the middle and bottom layer of blubber. Scale bar = 50 µm.

**Figure 5 animals-12-00537-f005:**
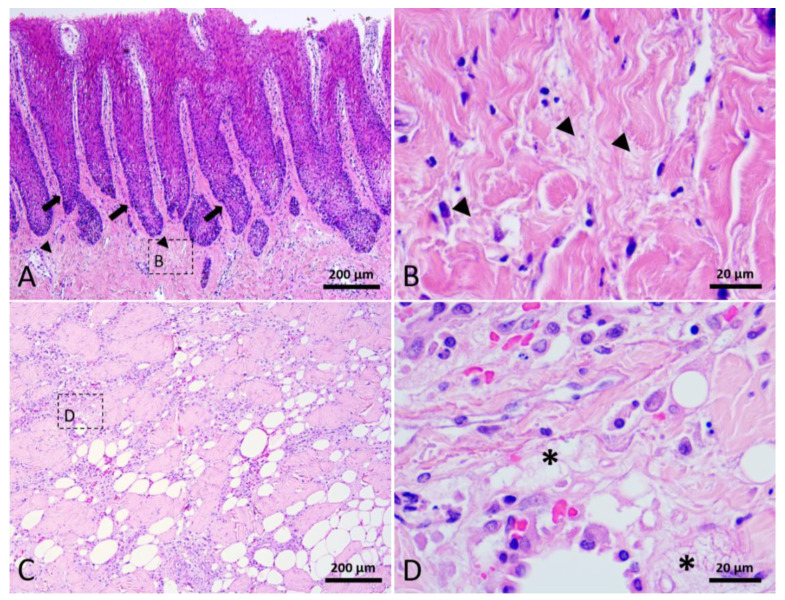
Stage 2 wound of Fraser’s dolphin. H&E staining of Fraser’s dolphin Stage 2 wound. (**A**) Low power view of Stage 2 wound epidermis and dermis. Degraded collagen fibers (arrowhead) in the dermis close to the wound edge. Black arrow: rete ridges. Scale bar = 200 µm. (**B**) High power view of degraded collagen fibers (arrowhead). Scale bar = 20 µm. (**C**) Low power view of blubber nearby the wound edge. Scale bar = 200 µm. (**D**) A loose fibrin network (asterisks) composed of fibrin, erythrocytes, and inflammatory cells replaced adipose tissue in the blubber. Scale bar = 20 µm.

**Figure 6 animals-12-00537-f006:**
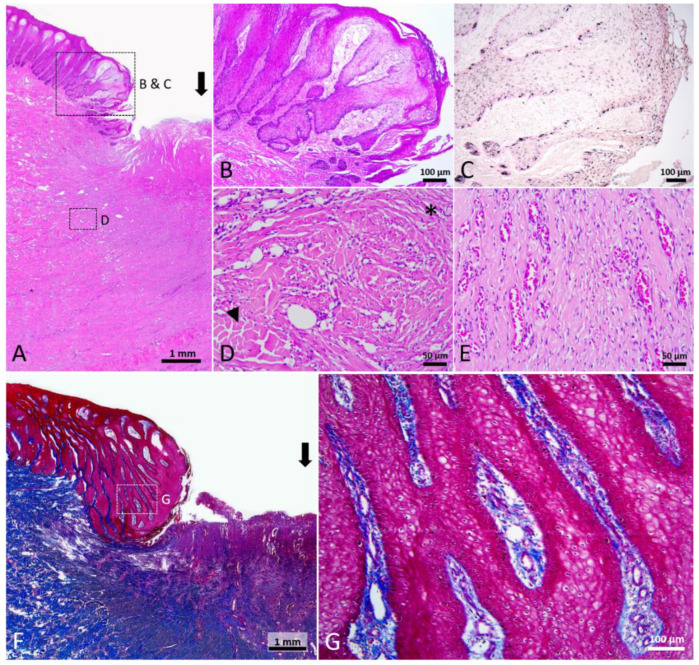
Stage 3 wound of Fraser’s dolphin. (**A**,**B**,**D**,**E**): H&E staining of Fraser’s dolphin Stage 3 wound. (**A**) Low power view of Stage 3 wound. Black arrow: wound center. Scale bar = 1 mm. (**B**) High power view of migrating epithelial tongue. Elongated epithelial strands appeared in the migrating epithelial tongue. Scale bar = 100 µm. (**C**) Fontana-Masson staining of Fraser’s dolphin Stage 3 wound. Melanocytes and melanin granules were present in the neo-epidermis. Scale bar = 100 µm. (**D**) High power view of Stage 3 wound dermis. Thick collagen fibers broke into pieces (arrowhead), and regions of adipose tissue were replaced by numerous cells and thin fibers (asterisks). Scale bar = 50 µm. (**E**) High power view of Stage 3 wound center. Wound center granulation tissue consisted of cells, extracellular matrix, and numerous microscopic blood vessels. Scale bar = 50 µm. (**F**,**G**): Trichrome staining of Fraser’s dolphin Stage 3 wound. (**F**) Low power view of Stage 3 wound. Block arrow: wound center. Scale bar = 1 mm. (**G**) High power view of migrating epithelial tongue. Neo-epidermis showed a reticular structure. Thin collagen fibers and blood vessels were present in the papillary dermis connected to the neo-epidermis. Scale bar = 100 µm.

**Figure 7 animals-12-00537-f007:**
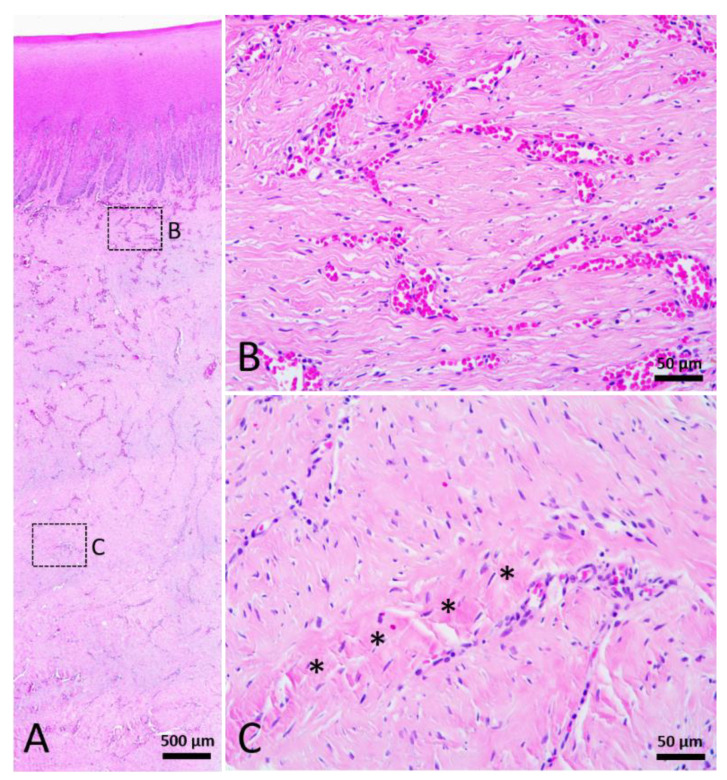
Stage 4 wound of Fraser’s dolphin. H&E staining of Fraser’s dolphin Stage 4 wound. (**A**) Low power view of Stage 4 wound. Scale bar = 500 µm. (**B**) High power view of Stage 4 wound dermis. Numerous blood vessels, cells, and thin collagen fibers existed in the reticular dermis of Stage 4 wounds. Scale bar = 50 µm. (**C**) High power view of Stage 4 wound blubber. There were a few thick collagen fibers (asterisks) in the lower part of the blubber. Scale bar = 50 µm.

**Figure 8 animals-12-00537-f008:**
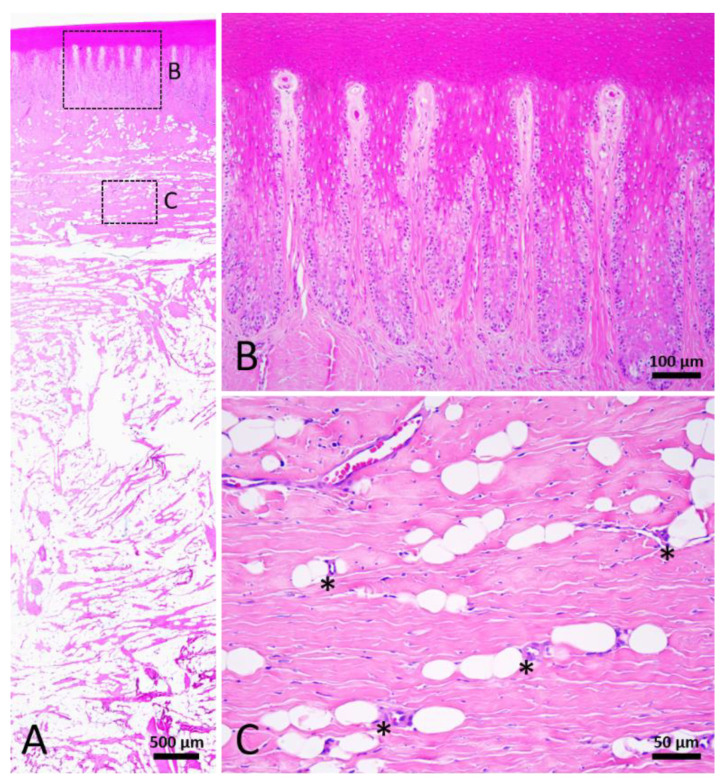
Stage 5 wound of Fraser’s dolphin. H&E staining of Fraser’s dolphin Stage 5 wound. (**A**) Low power view of Stage 5 wound. Normal skin structure including adipose tissue was noticed in the Stage 5 wound. Scale bar = 500 µm. (**B**) High power view of Stage 5 wound epidermis. Epidermis showed nearly normal rete and dermal ridges. Scale bar = 100 µm. (**C**) High power view of Stage 5 wound blubber. Adipocytes appeared in the perivascular region (asterisks). Scale bar = 50 µm.

**Table 1 animals-12-00537-t001:** Sample condition and the details of each animal used in the current study.

ID No.	TD20181128	TP20190115	IL20191105	ML20200807
**Species**	Fraser’s dolphin	Fraser’s dolphin	Fraser’s dolphin	Fraser’s dolphin
**Gender**	Male	Male	Male	Male
**Age**	Adult	Adult	Subadult	Adult
**Body length**	250 cm	247 cm	221 cm	244 cm
**Body condition**	Normal	Thin	Thin	Thin
**Carcass condition**	Code 2 (Freshly dead)	Code 2 (Freshly dead)	Code 2 (Freshly dead)	Code 3 (Moderate decomposition)
**Note**		Systemic infection (*Staphylococcus aureus)*		

**Table 2 animals-12-00537-t002:** Number of collected wound samples and macroscopic and microscopic features in different healing stages.

	Stage 1	Stage 2	Stage 3	Stage 4	Stage 5
**Number of samples**	2	6	1	14	13
**Macroscopic features**	Sharp edge; little or no bleeding; underlying tissue exposed	Raised and puckered epidermal edge; yellow-white substance on the surface of wound	Neo-epidermis in the wound margin; reddish tissue in the wound center	Complete closure; partial repigmentation	Little to no contraction lines; pigmentation similar to surrounding unwounded skin
**Microscopic features**					
**Epidermal changes**	Intercellular and intracellular edema; necrotic keratinocytes	Exfoliation; edema; necrosis	Melanocytes, melanin, rete and dermal ridges in the migrating epithelial tongue	Melanocytes, melanin, rete and dermal ridges in the neo-epidermis	Melanocytes, melanin, rete and dermal ridges in the epidermis
**Dermal changes**	No obvious change	Collagen degeneration	Wounded area: angiogenesis; granulation tissue formation. Adjacent area: numerous fibrocytes, fibroblasts, and thin collagen fibers.	Numerous fibrocytes, fibroblasts, blood vessels and thin collagen fibers	Few cells and blood vessels; thin/thick collagen fibers
**Blubber changes**	No obvious change	Collagen and adipose tissue necrosis; inflammatory cell infiltration; collagen degeneration; hyperemia/hemorrhage	Wounded area: angiogenesis; granulation tissue formation. Adjacent area: numerous fibrocytes, fibroblasts, and thin collagen fibers.	Numerous fibrocytes, fibroblasts, blood vessels and thin collagen fibers; limited adipose tissue	Few cells and blood vessels; thick collagen fibers; expanded adipose tissue.

## Data Availability

Not applicable.

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
