# Peer review of "Defining Wound Healing Progression in Cetacean Skin: Characteristics of Full-Thickness Wound Healing in Fraser’s Dolphins (Lagenodelphis hosei)"

_animals, 2022, doi:10.3390/ani12050537_

Round 1
Reviewer 1 Report
The authors provided an interesting, new attempt to investigate the wound in cetaceans, also proposing numerous, significant figures. However, some modifications must be introduced. Please, find some comments below and some minor ones in the pdf file.
- The M&M needs to be improved, for instance information on measurement of wound size, withdrawal of skin portion, etc. should be introduced.
- Not all the figures are referred to within the main text, e.g. about the pane of Fig. 6 only Figure 6B are introduced in the result paragraph.
- The pictures “A” of each figures may be presented in a single figure, which displays the evolution of stage. This means a sequence showing each stage, next to the next. E.g.
PICTURE OF STAGE 1-->PICTURE OF STAGE 2-->PICTURE OF STAGE 3 --> …..
Later, the figures "A" could be left in the panel showing the stage, maybe reduced in size.
- L 279-282: Move at the beginning of the paragraph 3.2. (the Table 2 too). It would be interesting also a brief description of the “macroscopic” features of each stage, as briefly showed in Table 2. At least, differences in size, colour, etc. among stages.
- In my mind, the identification of 5 stages of wound healing is one of the main results of this research. The authors should best explicate this concept, as well as highlight the modification/appearance/disappearance of the histo-morphological structures, which determine the “movement” form one stage to the next one.
- The authors should highlighted or, at least, propose some hypotheses about the possible effect of different carcass condition and presence of systemic infection on the wound. It is clear the few observation can not provide sure, definitive results, however some preliminary hypotheses may be proposed.
- Finally, I propose the authors highlight the effort to improve the knowledge on such subject, still not deeply explored. Indeed, the discussion section proposes several sparks for further studies and/or still needing investigation. It is true that numerous gaps of knowledge must be filled, however the present paper represent a step in this direction and such concept should be highlighted more than the next ones still needing to do.

Author Response
We would like to take this opportunity to express our sincere thanks to you for identifying areas of our manuscript that needed corrections or modification. Please see the attachment. We combined the comment reply and unformatted Word file with line numbers into a PDF.

Reviewer 2 Report
- The authors must clarify the meaning of "code 2" and "code 3" related to carcass condition in table 1 and possibly declare it; similarly for "normal" and "thin" related to body condition.
- In the caption of fig. 3 the same definition is used for "C" and "D" while it would be advisable to differentiate the two areas ( for example closer or further away from wound edge).
- In Fig 4 a lower power view should be inserted as in the other figures; in addition in the caption of fig. 4 enlargements A1 and B1 have not been inserted.
1. What is the main question addressed by the research? 2. Do you consider the topic original or relevant in the field, and if so, why? 3. What does it add to the subject area compared with other published material? 4. What specific improvements could the authors consider regarding the methodology? 5. Are the conclusions consistent with the evidence and arguments presented and do they address the main question posed? 6. Are the references appropriate? 7. Please include any additional comments on the tables and figures.
!. the authors intend to explain the full-thickness wound healing in dolphin to highlight differences between cetaceans and terrestrian mammals; 2. Yes, because there are few publications on the subject and the results could contribute to the treatment of injuries in the medical and veterinary fields; 3. This study firstly characterize histological features of full-thickness wound healing in cetacea to demonstrate the adaption to aquatic environment; 4. The authors could extend the study to the mechanisms of angiogenesis, collagen reconstruction and adipocyte regeneration in the aforementioned lesions. 5. yes; 6. yes; 7. Comments on the tables and figures have already been made
Author Response

(The authors gave the same response as above.)

Reviewer 3 Report
Overall, the work is weel designed and written. I suggest you write a paragraph providing an explanation for the importance of this type of work and how it integrates the work in your lab. This work would benefit from more context.
Author Response

(The authors gave the same response as above.)
